# *Siraitia grosvenorii* Extract Attenuates Airway Inflammation in a Mouse Model of Respiratory Disease Induced by Particulate Matter 10 Plus Diesel Exhaust Particles

**DOI:** 10.3390/nu15194140

**Published:** 2023-09-25

**Authors:** Yoon-Young Sung, Misun Kim, Heung Joo Yuk, Seung-Hyung Kim, Won-Kyung Yang, Geum Duck Park, Kyung Seok Kim, Woo Jung Ham, Dong-Seon Kim

**Affiliations:** 1KM Science Research Division, Korea Institute of Oriental Medicine, 1672 Yuseongdae-ro, Yuseong-gu, Daejeon 34054, Republic of Korea; yysung@kiom.re.kr (Y.-Y.S.); misun210@kiom.re.kr (M.K.); yukhj@kiom.re.kr (H.J.Y.); 2Institute of Traditional Medicine and Bioscience, Daejeon University, 62 Daehak-ro, Dong-gu, Daejeon 34520, Republic of Korea; sksh518@dju.kr (S.-H.K.); ywks1220@dju.kr (W.-K.Y.); 3Suheung Research Center, Seongnam 13488, Republic of Korea; kdpark@suheung.com (G.D.P.); kskim1@suheung.com (K.S.K.); wjham@suheung.com (W.J.H.)

**Keywords:** diesel exhaust particles, neutrophil, NF-κB, particulate matter, inflammatory respiratory disease, *Siraitia grosvenorii*

## Abstract

Exposure to particulate matter (PM) causes considerable breathing-related health risks. *Siraitia grosvenorii* fruit is a traditional remedial plant used in Korea and China to treat respiratory diseases. Our recently published study showed that *S. grosvenorii* extract (SGE) ameliorated airway inflammation in lipopolysaccharide- and cigarette-smoke-induced chronic obstructive pulmonary disease in mice. Thus, we aimed to assess the inhibitory effects of SGE on airway inflammation in mice exposed to a fine dust mixture of PM10 (PM diameter < 10 mm) and diesel exhaust particles (DEPs) known as PM10D. The mice (BALB/c) were treated with PM10D via intranasal injection three times over a period of 12 days, and SGE 70% ethanolic extract (50 or 100 mg/kg) was orally administered daily for 12 days. SGE attenuated neutrophil accumulation and the number of immune B and T cells from the lung tissue and bronchoalveolar lavage fluid (BALF) of the PM10D-exposed mice. SGE reduced the secretion of cytokines and chemokines, including interleukin (IL)-1α, tumor necrosis factor (TNF)-α, IL-17, C-X-C motif chemokine ligand (CXCL)1, and macrophage inflammatory protein (MIP)-2 in the BALF. Airway inflammation, infiltration of inflammatory cells, and collagen fibrosis in the lung after PM10D exposure were investigated via histopathological analysis, and SGE treatment ameliorated these symptoms. SGE decreased the mRNA expression of mucin 5AC (MUC5AC), CXCL1, TNF-α, MIP-2, and transient receptor potential ion channels in the lung tissues. Furthermore, SGE ameliorated the activation of mitogen-activated protein kinase (MAPK)/nuclear factor-kappa B (NF-κB) signaling by PM10D in the lungs. We conclude that SGE attenuated PM10D-induced neutrophilic airway inflammation by inhibiting MAPK/NF-κB activation. These results show that SGE may be a candidate for the treatment of inflammatory respiratory diseases.

## 1. Introduction

Particulate matter (PM) is suspended dust, with each particle having a diameter of less than 10 μm (PM10), and can be inhaled and accumulate in lung alveoli, causing considerable respiratory health risks [1]. Exposure to PM causes aggravated respiratory symptoms, induces pulmonary inflammation, and exacerbates allergic responses, owing to its stimulatory effect on oxidative stress and its toxicity, leading to a rise in the prevalence of bronchial asthma, bronchitis, emphysema, respiratory infection, neurodegenerative diseases, cardiovascular diseases, chronic respiratory disorder, and lung cancer [2,3,4,5]. In Asia, PM contains seasonal yellow dust, fine dust from China, and domestic air pollutants generated from combustion sources, such as gasoline and diesel engines, coal combustion, and biomass boiling. In particular, diesel exhaust particles (DEPs) consist of hundreds of different chemicals (primarily carbon, polycyclic aromatic hydrocarbon, ash, metallic abrasion particles, sulfates, nitrates, and other trace elements) and lead to the highest toxicity in the human respiratory system [6,7]. Exposure to PM and DEPs leads to the accumulation of these particulates in the alveolar region (epithelial cell and alveolar macrophage) of the lungs, eventually triggering airway inflammation by impaired immune responses in the lung and airway to induce or worsen respiratory disorders, such as chronic obstructive pulmonary disease (COPD) and allergic asthma [8]. Therefore, there is a need to develop novel therapeutic candidates against PM-induced respiratory damage for human respiratory health.

*Siraitia grosvenorii* Swingle fruit (also known as monk fruit or Luo Han Guo, of the Cucurbitaceae family) is a traditional remedial plant used in Korea and Southern China to treat respiratory diseases, such as cold, cough, laryngitis, bronchitis, and sore throat, as well as stomach ailments, and is also used as a low-calorie sweetener in food and drinks [9]. The various biological effects of *S. grosvenorii* extract and its major compounds mogrosides have been studied. *S. grosvenorii* extract (SGE) and mogrosides showed antiglycative, antioxidative, anti-microbial, anti-inflammatory, anti-asthmatic, anti-diabetic, hypoglycemic, anti-tussive, anti-carcinogenic, anti-viral, hepatoprotective, neuroprotective, and immunomodulatory properties [10,11,12,13]. In addition, SGE alleviated symptoms of allergic illnesses, such as asthma and atopic dermatitis, via inhibiting allergic inflammation of the lungs and skin in ovalbumin- and mite-allergen-exposed mice [14,15]. As the main components of *S. grosvenorii* fruit, mogroside V and 11–oxomogroside V as well as SGE decreased airway hyperresponsiveness and airway inflammatory response in the ovalbumin-inhaled asthmatic mice via preventing nuclear factor-kappa B (NF-κB) activation and acetylcholine binding to airway muscarinic-3 acetylcholine receptor (M_3_R) [9,16]. Our recently published study showed that *S. grosvenorii* ethanolic extract ameliorated airway inflammation in lipopolysaccharide-stimulated bronchial epithelial cells and in lipopolysaccharide- and cigarette-smoke-induced COPD mice [17]. These studies suggest that SGE may be a candidate for human respiratory health. However, there are no studies about the improvement effect of SGE on airway inflammation when exposed to PM10 or DEP. Therefore, the current study aimed to examine the effect of SGE on airway inflammation in a fine dust mixture of PM10 and DEP (PM10D)-exposed chronic respiratory disorder in a mouse model and the signaling mechanism underlying its effects. These findings suggest, for the first time, that *S. grosvenorii* improves fine-dust-induced respiratory damage.

## 2. Materials and Methods

### 2.1. Extraction of SGE

The *S. grosvenorii* extract, Mogron^®^, was given to us by Suheung Co., Ltd. (Cheongju, Republic of Korea). Its preparation has been described previously [15]. Briefly, the dried fruit of *S. grosvenorii* was extracted with water. After water extraction, *S. grosvenorii* residues were dried and extracted with 70% (*v*/*v*) ethanol.

### 2.2. Mouse Model

Male BALB/c mice (Seven-week-old), bought from Orient Bio Inc. (Seongnam, Republic of Korea), were kept in animal facility (Laboratory Animal Research Center of Daejeon University) at a temperature of 21 °C ± 2 °C and a humidity of 60% ± 10%. Approval for the studies was given by the Committee for Animal Welfare at Daejeon University (ethical approval code: DJUARB2022-041), and the experiment was performed in accordance with the *Guide for the Care and Use of Laboratory Animals*. After acclimatization, the mice were randomized into five groups (*n* = 8 mice for each group): (i) vehicle-treated normal control (NC), (ii) PM10D control (CTL), (iii) PM10D-dexamethasone 3 mg/kg, (iv) PM10D-SGE 100 mg/kg, or (v) PM10D-SGE 50 mg/kg. The dose of SGE was decided based on our prior studies using a COPD mouse model [15]. PM10 and DEP were dissolved in 99% saline and 1% aluminum-hydroxide-based gel adjuvants. Chronic inflammation was induced by intranasal administration of a fine dust mixture (PM10D), which included 3 mg/mL PM10 (ERMCZ120, MilliporeSigma, Burlington, MA, USA) and 0.6 mg/mL DEP (NIST2975, MilliporeSigma), to the mice on days 4, 7, and 10, as reported previously [16]. The vehicle solution was given to the normal control mice intranasally. The fine dust mixture was given to the remaining mice intranasally. Dexamethasone (3 mg/kg) as a positive control or SGE (50 or 100 mg/kg) was orally administered every day for 12 days. The experimental setup is expressed in Figure 1A. On day 12, all mice were euthanized using anesthesia with zoletil and blood, bronchoalveolar lavage fluid, and lung tissues collected for further experiments. Lung tissues were analyzed using different techniques (right lung for histopathology; left lung for flow cytometry, Western blot, and quantitative reverse-transcription–polymerase chain reaction).

### 2.3. Collection of Mouse Lung Cells from Bronchoalveolar Lavage Fluid

Bronchoalveolar lavage fluid (BALF) was acquired on day 12 by cannulating the trachea of the mice and adding Dulbecco’s modified Eagle medium into the lung, and the medium was then collected. The number of lung cells collected from the BALF was counted, and the cells were collected using cytospin (Hanil, Gimpo, Republic of Korea) and stained with a Diff-Quik solution. The differential cell count was evaluated using a cytospin slide. The lung cells were separated as a single-cell suspension and treated in phosphate-buffered saline containing collagenase IV (1 mg/mL, MilliporeSigma) for 35 min at 37 °C. Then, the filtered cell suspension was centrifuged at 492× *g* for 20 min, and the pellets were collected.

### 2.4. Measurement of Cytokines

Interleukin (IL)-1α, IL-17, C-X-C motif chemokine ligand (CXCL)1, macrophage inflammatory protein (MIP)-2, and tumor necrosis factor (TNF)-α levels from the BALF were determined via enzyme-linked immunosorbent assay kits (R&D Systems, Minneapolis, MN, USA). The optical density was determined at 450 nm using a SpectraMax analyzer (Molecular Devices, San Jose, CA, USA).

### 2.5. Histopathological Examination of Lung Tissue

The mouse lung (right) was perfused with 1 mL 10% (*v*/*v*) neutral-buffered formalin fixation solution through the trachea; then, the lungs were removed and submerged in formalin buffer. The tissue was paraffinized, cut to 5 μm thickness, and stained with Masson’s trichrome (MT) or hematoxylin and eosin (H&E) solution for the observation of inflammatory cell penetration and collagen fiber formation (Sigma-Aldrich, Seoul, Republic of Korea). The severity degree of inflammation was scored in a double-blind manner by two independent researchers and was determined using a subjective scale of 0–2 described by Lee et al. [8].

### 2.6. Quantitative Reverse-Transcription–Polymerase Chain Reaction

The lung tissue was purified for total RNA using a total RNA prep kit (HiGene, BIOFACT, Daejeon, Republic of Korea). To quantify the mRNA expression of the genes, quantitative reverse-transcription–polymerase chain reaction (qRT-PCR) was completed using an Applied Biosystems 7500 Fast (Thermo Fisher Scientific, Waltham, MA, USA) with an SYBR Green master mixture (Applied Biosystems) and primer. The gene transcript was expressed as ΔΔCt and normalized to the *β-actin.* The oligonucleotide sequences of primer are recorded in Table 1.

### 2.7. Flow Cytometry Analysis

The cells were collected from the lungs and BALF and incubated for 30 min with antibodies against cluster of differentiation (CD)8a (Clone 53-6.7, Isotype rat IgG2a), CD69 (Clone H1.2F3, Isotype hamster IgG1), CD4 (Clone RM4-5, Isotype rat IgG2a), CD62L (Clone MEL-14, Isotype rat IgG2a), CD44 (Clone IM7, Isotype rat IgG2b), CD21/CD35 (Clone 7G6, Isotype rat IgG2b), sialic-acid-binding immunoglobulin-like lectin F (SiglecF, Clone 1RNM44N, Isotype rat IgG2a), B220 (Clone RA3-6B2, Isotype rat IgG2a), granulocytes (Gr)-1 (Clone RB6-8C5, Isotype rat IgG2b), and CD11b (Clone M1/70, Isotype rat IgG2b). All antibodies were obtained from BD Biosciences (Franklin Lakes, NJ, USA), except for SiglecF antibody (Thermo Fisher Scientific). The cells were reacted with phycoerythrin- or fluorescein-isothiocyanate-labeled secondary antibodies for 35 min, washed, and fixed with 4% paraformaldehyde (Sigma-Aldrich) for 25 min. The cells were analyzed on a FACSCalibur (BD Biosciences).

### 2.8. Immunoblot

Proteins from the lung tissues were extracted in protein extraction buffer (PRO-PREP, Intron Biotechnology, Seongnam, Republic of Korea), divided through gel electrophoresis, and transferred to a membrane by a Trans-Blot Turbo transfer system (Bio-Rad Laboratories, Hercules, CA, USA). Equal amounts of proteins (5~17 μg) were loaded into each lane. After electrophoresis, whole protein amount was verified using a total protein stain by Coomassie Blue. The blocked membrane was performed using blocking solution (EzBlockChemi, ATTO, Daejeon, Republic of Korea) for 40 min. Then, the membrane was treated with phosphorylated nuclear factor kappa light-chain enhancer of activated B cells (NF-κB), nuclear factor of kappa light polypeptide gene enhancer in B-cells inhibitor (IκB), c-jun NH2 terminal kinase (JNK), p38 mitogen-activated protein kinase (MAPK), extracellular signal-regulated kinase (ERK), NF-κB, ERK, IκB, p38, or JNK (Cell Signaling Technology, Danvers, MA, USA) antibodies. The proteins were analyzed relative to β-actin at the end. All antibodies were diluted at a 1:1000 ratio. The membrane was incubated with anti-rabbit secondary antibodies (Cell Signaling, 1:10,000) for 40 min. The signal was obtained with chemiluminescence (Thermo Fisher Scientific, Waltham, MA, USA). Images of the membranes were evaluated in ImageJ version 1.52a software.

### 2.9. Statistical Analysis

The results are expressed as the mean ± standard error of the mean. Statistical testing among groups was performed using one-way analysis of variance and Duncan’s multiple comparison test. The statistical analysis was performed using GraphPad Prism 7.0 software. *p* < 0.05 was considered to demonstrate a significant difference. # *p* < 0.05, ## *p* < 0.01, and ### *p* < 0.001 were compared with the normal control group, and * *p* < 0.05, ** *p* < 0.01, and *** *p* < 0.001 were compared with the PM10D control group.

## 3. Results

### 3.1. Effect of SGE on Neutrophil Infiltration in PM10D-Exposed Mice

The exposure of the mice to the fine dust mixture (PM10D) for 12 days increased the total cell number in the BALF and lungs (*p* = 0.0009) (Figure 1B,C). The total cell number in the BALF decreased after oral administration of SGE (100 mg/kg, *p* = 0.0009; 50 mg/kg, *p* = 0.048) or dexamethasone (*p* = 0.049), but the total number of lung cells did not (*p* = 0.393) (Figure 1B,C). In particular, the number of neutrophils in the BALF by cytospin elevated after PM10D exposure compared with the NC mice (*p* = 0.0003), and neutrophil infiltration significantly decreased after SGE administration (*p* = 0.0007) (Figure 1D,E).

### 3.2. Effect of SGE on White Blood Cells in the Blood

Changes in the cell types in the blood, particularly neutrophils, offer significant information about the development of airway inflammation. Hematological examination of white blood cells (WBCs) in the blood indicated that the total count of WBCs decreased after PM10D exposure compared with the NC (*p* = 0.0009), and the decreased levels recovered after SGE administration (*p* = 0.0007) (Figure 2A). The number of neutrophils increased after PM10D exposure compared with the NC (*p* = 0.00001) and significantly decreased after SGE administration (*p* = 0.001) (Figure 2B). The number of lymphocytes decreased after PM10D exposure compared with the NC (*p* = 0.0009) and increased after SGE administration (*p* = 0.00004) (Figure 2B). Monocytes, eosinophils, and basophils did not demonstrate significant differences among the groups (*p* > 0.05) (Figure 2C).

### 3.3. Effect of SGE on the Release of Inflammatory Mediators in BALF

Inflammatory chemokines and cytokines secreted by the inflammatory response contribute to the pathology and severity of airway inflammation. The IL-1α, TNF-α, CXCL1, MIP-2, and IL-17 levels in the BALF increased after PM10D exposure compared with the NC (*p* < 0.01) and were significantly suppressed following the administration of SGE (IL-1α, *p* = 0.017; TNF-α, *p* = 0.049; CXCL1, *p* = 0.0019; MIP-2, *p* = 0.0001; IL-17, *p* = 0.0049) or dexamethasone (IL-1α, *p* = 0.0071; TNF-α, *p* = 0.049; CXCL1, *p* = 0.0009; MIP-2, *p* = 0.0002; IL-17 *p* = 0.0009) (Figure 3A–E).

### 3.4. Lung Histopathology

To examine the effect of SGE on lung histopathology in PM10D-induced airway inflammation, the lung tissue was dyed with Masson’s trichrome or hematoxylin and eosin stain solutions. We observed thickening of the airway wall with the infiltration of inflammatory cells around the airway and collagen fibrosis in the lungs of the PM10D-exposed mice (*p* = 0.00023). In contrast, lung sections from mice administrated with SGE or dexamethasone had reduced inflammatory cell infiltration and fibrosis (*p* = 0.0009) (Figure 4A–C). These results show that SGE constrains histopathological changes to airway inflammation in the lungs of PM10D-exposed mice.

### 3.5. Effect of SGE on the Expression of Inflammatory Mediators in the Lung Tissue

As shown in Figure 5A–F, the mRNA expression of *mucin 5AC (MUC5AC)*, *CXCL1*, *transient receptor potential (TRP) vanilloid 1 (TRPV1)*, *TRP ankyrin 1 (TRPA1)*, *MIP-2*, and *TNF-α* in the lung increased after PM10D exposure compared with the NC mice (*p* < 0.01) and was significantly inhibited after SGE administration (*MUC5AC*, *p* = 0.00002; *CXCL1*, *p* = 0.018; *TRPA1*, *p* = 0.024; *TRPV1*, *p* = 0.042; *MIP-2*, *p* = 0.0074; *TNF-α*, *p* = 0.0076) or dexamethasone (*MUC5AC*, *p* = 0.00005; *CXCL1*, *p* = 0.0051; *TRPA1*, *p* = 0.021; *TRPV1*, *p* = −0.143; *MIP-2*, *p* = 0.0019; *TNF-α*, *p* = 0.0027) (Figure 5).

### 3.6. Effect of SGE on Immune Cell Numbers in the Lung Tissue and BALF

The effects of SGE on the change in immune cell numbers after PM10D exposure were examined through flow cytometry analysis of the BALF and lung tissue. Neutrophil numbers in the BALF and lungs increased after PM10D exposure (*p* = 0.0009) and decreased after SGE (100 mg/kg) administration (*p* = 0.0009) (Table 2). The absolute number of activated T lymphocytes (CD8^+^, CD4^+^, and CD62L^−^/CD44^high+^) and neutrophils (Gr-1^+^/SiglecF^−^) in the BALF increased after PM10D exposure (*p* = 0.009, *p* = 0.0009, *p* = 0.0052, and *p* = 0.0056, respectively) and decreased after the administration of 100 mg/kg SGE (*p* = 0.009, *p* = 0.030, *p* = 0.0168, and *p* = 0.0097, respectively) or dexamethasone (*p* = 0.049, *p* = 0.0053, *p* = 0.1937, and *p* = 0.0210, respectively). Additionally, the absolute number of activated T cells (CD62L^−^/CD44^high+^ and CD4^+^CD69^+^), neutrophils (Gr-1^+^/SiglecF^−^), and myeloid cells (GR-1^+^CD11b^+^) in the lungs increased after PM10D exposure (*p* = 0.0009, *p* = 0.009, *p* = 0.0015, and *p* = 0.0083, respectively) and significantly decreased after the administration of 100 mg/kg SGE (*p* = 0.049, *p* = 0.049, *p* = 0.0021, and *p* = 0.0159, respectively) or dexamethasone (*p* = 0.0075, *p* = 0.049, *p* = 0.0043, and *p* = 0.0265, respectively). These results indicate that SGE suppressed the bronchial immune response and neutrophilic airway inflammation observed after PM10D exposure.

### 3.7. Effect of SGE on the MAPK/NF-κB Signaling Pathway in the Lungs

To find the potential active signaling pathways controlling the inhibitory effect of SGE on the airway inflammatory response in PM10D-exposed mice, we analyzed the activation of signaling molecules in the MAPK/NF-κB pathway (Figure 6, Appendix A). The activation of ERK (*p* = 0.0007), p38 (*p* = 0.0003), and JNK (*p* = 0.012) via phosphorylation increased after PM10D exposure and decreased after SGE (100 mg/kg, *p* = 0.0001, *p* = 0.017, and *p* = 0.0015, respectively; 50 mg/kg, *p* = 0.0001, *p* = 0.5956, and *p* = 0.0018, respectively) or dexamethasone administration (*p* = 0.0001, *p* = 0.0043, and *p* = 0.0030, respectively). The activation of NF-κB-p65 and IκB by phosphorylation was also upregulated after PM10D exposure (*p* = 0.0049 and *p* = 0.0060) and decreased after SGE administration (100 mg/kg, *p* = 0.0048 and *p* = 0.0031; 50 mg/kg, *p* = 0.0439 and *p* = 0.0055). This result shows that the anti-inflammatory activity of SGE on airway inflammation is connected to the MAPK/NF-κB (ERK, p38, and JNK) signaling pathway.

## 4. Discussion

In the current study, we observed the inhibitory activities of SGE on airway inflammation in a PM10D-induced chronic inflammatory disease mice model. The release of the inflammatory mediators IL-1α, TNF-α, CXCL1, IL-17, and MIP-2 in BALF was reduced by SGE administration. In addition, SGE inhibited the mRNA levels of *TRPA1*, *TRPV1*, *MUC5AC*, and inflammatory cytokine (*CXCL-1*, *MIP2*, and *TNF-α*) expression and also reduced histopathological changes, such as inflammatory cell infiltration and collagen fibrosis in the lungs of the PM10D mice. The neutrophil number in the WBC count, lung, and BALF was decreased effectively via SGE administration. The cell number of CD8+ T cells, CD62L^−^/CD44^high+-^activated T cells, CD4^+^ T cells, and neutrophils (Gr-1^+^/SiglecF^−^) from the BALF was suppressed via SGE administration, and the cell number of activated CD4^+^CD69^+^ and CD62L^−^/CD44^high+^ T cells, Gr-1^+^/SiglecF^−^ neutrophils, and myeloid-derived suppressor cells (GR-1^+^CD11b^+^) from the lung tissue was also decreased via SGE administration in the PM10D-exposed mice.

Neutrophilic airway inflammation is crucial in the early and progressive stages of pulmonary illness, and IL-1R signaling by IL-1α in the early stages drives neutrophilic inflammation and subsequent structural changes in the lungs and bronchi [18]. Thus, the increased levels of IL-1α in the BALF of the PM10D-exposed mice observed in the current study were correlated with neutrophilic inflammation. The predominant activated T-cell subtypes, such as CD8^+^ and CD4^+^ T cells, lead to airway neutrophilic inflammation through the secretion of proinflammatory cytokines [19]. The CD4^+^ T cells cause the aggravation of chronic airway inflammation by producing IL-17 [20]. The granulocyte indicator Gr-1 is associated with the differentiation and maturation of granulocytes, and CD11b is an indicator of myeloid cells of the macrophage lineage [21,22]. Thus, the population of Gr-1^+^CD11b^+^ cells with Gr-1^+^/SiglecF^−^ cells may constitute a substantial portion of neutrophils after PM10D exposure [23,24].

Chemokines, such as CXCL-1 and MIP2 (CXCL-2), are major neutrophil chemoattractants that are produced in the lung in an airway inflammation model induced by DEP exposure; an increase in the pulmonary expression of these C-X-C chemokines exacerbated airway inflammation [25]. As a key constituent of respiratory mucin secreted in the bronchial epithelia, MUC5AC contributes to airway mucus hypersecretion in respiratory illnesses, such as asthma and COPD [26,27]. Both TRPV1 and TRPA1, members of the TRP channel superfamily, play a key part in lung inflammation via the release of neuropeptides (substance P) and proinflammatory factors, such as leukotriene, IL-1α, and TNF-α, which cause primary airway inflammation [28]. Activation of these TRP channels by exposure to environmental lung toxic irritants, such as PM, DEP, and cigarette smoke, causes coughing by stimulation of nociceptive C-fibers in the airways of humans and animals, and TRP-channel-induced neurogenic inflammation could lead to the progression and symptoms (cough reflex) of airway inflammatory illnesses, such as lung fibrosis, COPD, and allergic asthma [23,28,29,30].

This study has shown that the increased cell number of immune cells, such as neutrophils, T cells (CD4^+^CD69^+^, CD8^+^, CD62L^−^/CD44^high+^, CD4^+^), Gr-1^+^/SiglecF^−^ neutrophils, and myeloid-derived suppressor cells (GR-1^+^CD11b^+^), in BALF and lung tissue by PM10D exposure was reduced via SGE administration in mice. In addition, increased levels of expression of inflammatory mediators, such as CXCL1, MIP-2, MUC5AC, TRPV1, TRPA1, IL-1α, TNF-α, and IL-17 in BALF and lung were reduced by SGE administration. These results indicate that SGE effectively improves PM10D-induced neutrophilic airway inflammation, respiratory damage (fibrosis), and typical respiratory symptoms (mucus secretion and cough) by inhibiting the immune cells and inflammatory mediators.

Exposure to environmental hazards, such as PM, DEP, and cigarette smoke, stimulates proinflammatory mediators, such as IL-1 and TNF-α, which activate the NF-κB transcription factor or MAPK signaling molecules. This process leads to lung inflammation with neutrophil recruitment to the lung via the pulmonary expression of cytokines and neutrophil chemokines such as CXCL-1 and MIP-2 [7,31,32]. Consistent with previous reports, our results show that SGE alleviates neutrophilic airway inflammation by preventing NF-κB and ERK/p38/JNK MAPK signaling (Figure 7).

The maximum effective dose of SGE for the PM10D-induced chronic inflammatory disease mice in this study is 100 mg/kg. The practical human dose of SGE based on body surface area is about 500 mg/day (60 kg adult, 1–2 times/day). However, the appropriate dosage for human administration can differ depending on frequency, gender, and age. A limitation of the current study is that it did not test the effects of SGE in both sexes of mice. Recently, studies have reported that in humans, females are more susceptible than males to airway inflammation caused by environmental air pollution owing to sex hormone differences [33]. Thus, the effect of SGE in female mice should be tested in a future study. Another limitation of this study is that there is no lung function (pulmonary function tests that check how well the lungs work) and airway morphometry data. Nevertheless, our studies show that SGE could be a potent candidate to prevent respiratory illnesses by ameliorating the airway inflammation caused by PM10D. These discoveries reveal, for the first time, that *S. grosvenorii* ameliorates fine-dust-induced respiratory damage. From the current results, a human clinical trial on SGE for the development of human respiratory health food will be conducted in the future.

## 5. Conclusions

This study has shown that SGE ameliorates neutrophilic airway inflammation and lung injury through the downregulation of NF-κB and MAPK signaling pathways in a PM10D-induced respiratory disease murine model. These results suggest that *S. grosvenorii* would be a potent candidate for the management of respiratory disorders.

## Figures and Tables

**Figure 1 nutrients-15-04140-f001:**
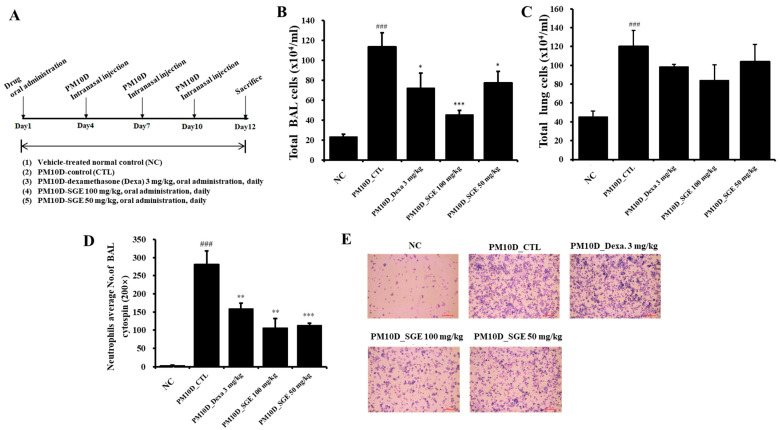
Experimental setup and effect of *Siraitia grosvenorii* extract (SGE) on total and immune cell numbers in a PM10D-induced airway inflammation model. (**A**) Experimental setup. Number of (**B**) total bronchoalveolar lavage fluid (BALF) cells, (**C**) total lung cells, and (**D**) neutrophils in (**E**) BALF cytospin (magnification: 200×). *n* = 8/group. ### *p* < 0.001 vs. normal control (NC) mice. * *p* < 0.05, ** *p* < 0.01, and *** *p* < 0.001 vs. PM10D control (CTL) mice.

**Figure 2 nutrients-15-04140-f002:**
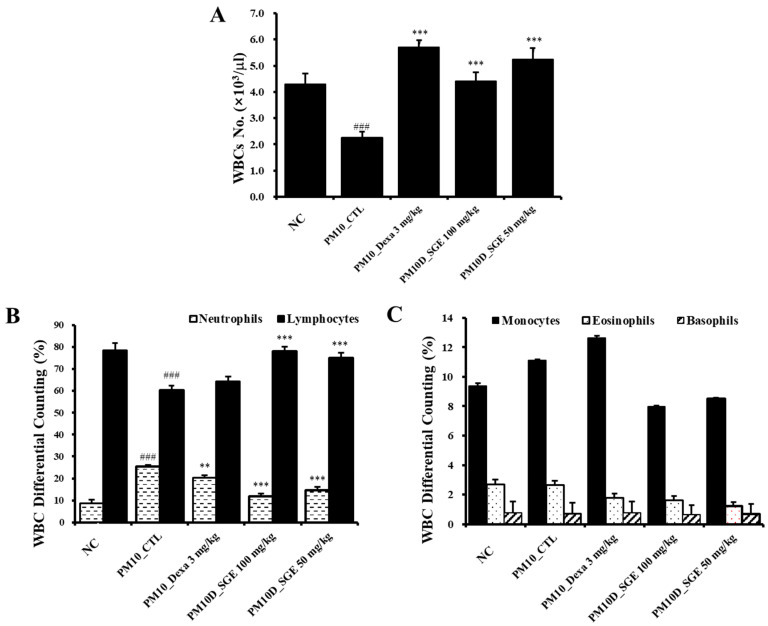
Effects of *Siraitia grosvenorii* extract (SGE) on (**A**) the number of white blood cells (WBCs) and (**B**,**C**) WBC differential cells counting. *n* = 8. ### *p* < 0.001 vs. the normal control (NC) mice. ** *p* < 0.01, and *** *p* < 0.001 vs. the PM10D control (CTL) mice.

**Figure 3 nutrients-15-04140-f003:**
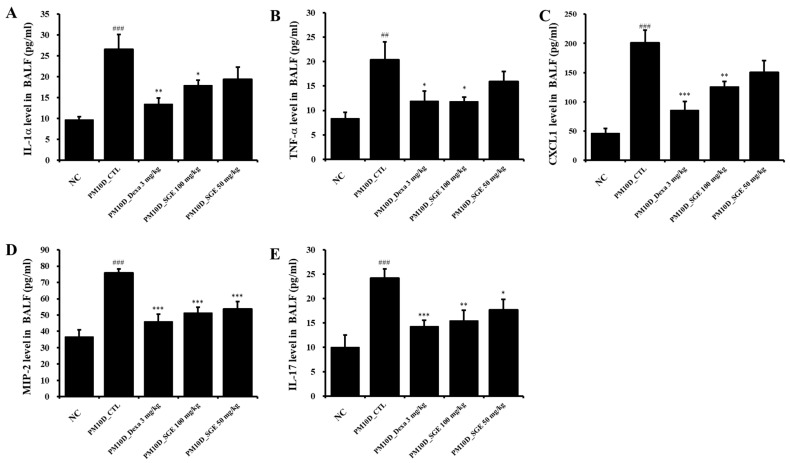
Effect of SGE on the release of cytokines and chemokines in the bronchoalveolar lavage fluid (BALF). BALF production of (**A**) IL-1α, (**B**) TNF-α, (**C**) CXCL1, (**D**) MIP2, and (**E**) IL-17 (*n* = 8/group). ## *p* < 0.01, and ### *p* < 0.001 vs. the normal control (NC) mice. * *p* < 0.05, ** *p* < 0.01, and *** *p* < 0.001 vs. the PM10D control (CTL) mice.

**Figure 4 nutrients-15-04140-f004:**
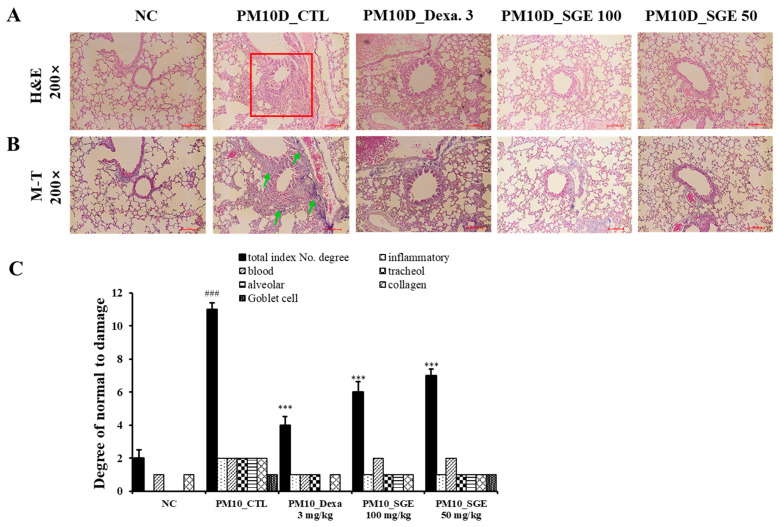
Lung histopathology. (**A**) Hematoxylin and eosin (H&E) staining and (**B**) Masson’s trichrome (MT) staining of the lung tissue (magnification: 200×). (**C**) Semi-quantitative analysis of the degree of lung damage (*n* = 8/group). ### *p* < 0.001 vs. the normal control (NC) mice. *** *p* < 0.001 vs. the PM10D control (CTL) mice.

**Figure 5 nutrients-15-04140-f005:**
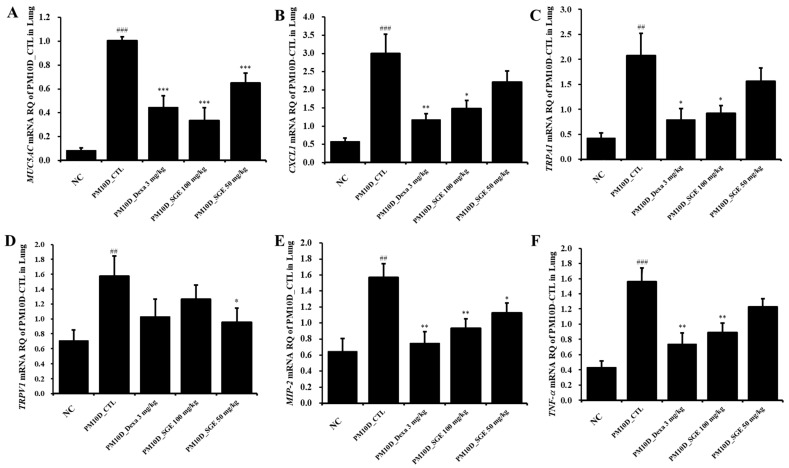
Effects of *Siraitia grosvenorii* extract (SGE) on the mRNA expression of airway-inflammation-related genes in the lungs of mice. The mRNA levels of (**A**) *MUC5AC*, (**B**) *CXCL1*, (**C**) *TRPA1*, (**D**) *TRPV1*, (**E**) *MIP-2*, and (**F**) *TNF-α* (*n* = 8/group). ## *p* < 0.01, and ### *p* < 0.001 vs. the normal control (NC) mice. * *p* < 0.05, ** *p* < 0.01, and *** *p* < 0.001 vs. the PM10D control (CTL) mice.

**Figure 6 nutrients-15-04140-f006:**
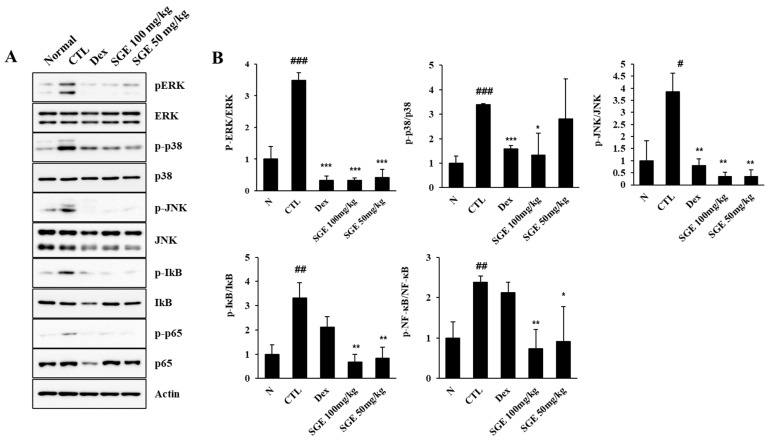
Effects of SGE on PM10D-treated mitogen-activated protein kinase (MAPK)/nuclear factor-kappa B (NF-κB) signaling in the lung tissue. (**A**) The protein expression of β-actin, p-p65, p65, pERK, ERK, p-JNK, JNK, p-p38, and p38 and (**B**) quantitative analysis of protein bands were evaluated with ImageJ (*n* = 8/group). # *p* < 0.05, ## *p* < 0.01, and ### *p* < 0.001 vs. the normal control (NC) mice. * *p* < 0.05, ** *p* < 0.01, and *** *p* < 0.001 vs. the PM10D control (CTL) mice.

**Figure 7 nutrients-15-04140-f007:**
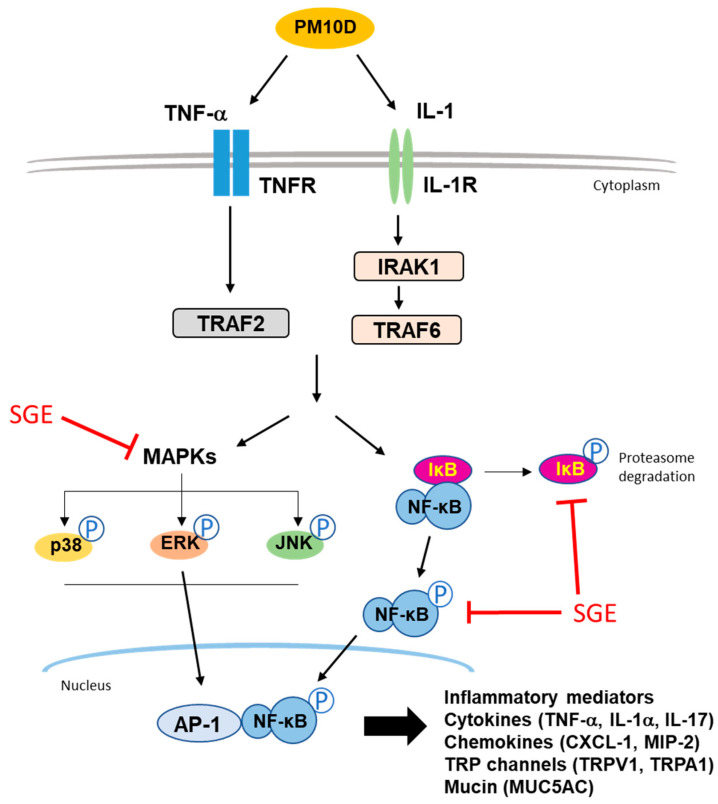
Schematic summary of the signaling pathways affected by SGE.

**Table 1 nutrients-15-04140-t001:** Oligonucleotide sequences of primer used for qRT-PCR.

Gene	Primer Direction	Oligonucleotide Sequence (5′-3′)
*β-actin*	F	TGGAATCCTGTGGCATCCAT
	R	TAAAACGCAGCTCGTAACAG
*TNF-α*	F	CCTGTAGCCCACGTCGTAGC
	R	TTGACCTCAGCGCTGAGTTG
*MIP-2*	F	ATGCCTGAAGACCCTGCCAAG
	R	GGTCAGTTAGCCTTGCCTTTG
*CXCL1*	F	CCGAAGTCATAGCCACAC
	R	GTGCCATCAGAGCAGTCT
*MUC5AC*	F	AGAATATCTTTCAGGACCCCT
	R	ACACCAGTGCTGAGCATACTT
*TRPV1*	F	CATCTTCACCACGGCTGCTTAC
	R	CAGACAGGATCTCTCCAGTGAC
*TRPA1*	F	TGAGATCGACCGGAGT
	R	TGCTGAAGGCATCTTG

Abbreviations: F, forward; R, reverse; *MIP-2*, macrophage inflammatory protein 2; *TNF-α,* tumor necrosis factor-*α*; *CXCL1*, C-X-C motif chemokine ligand-1; *MUC5AC*, mucin 5AC; *TRPV1*, transient receptor potential vanilloid 1; *TRPA1*, transient receptor potential ankyrin 1.

**Table 2 nutrients-15-04140-t002:** Effects of *Siraitia grosvenorii* extract (SGE) on immune cell subtypes in the bronchoalveolar lavage fluid (BALF) and lungs using flow cytometry analysis.

Cell Types	Absolute No.				
	NC	PM10D-CTL	PM10D-Dexa 3 3 mg/kg	PM10D-SGE 100 mg/kg	PM10D-SGE 50 mg/kg
BALF					
Lymphocytes (×10^4^ cells)	2.73 ± 0.68	7.45 ± 1.76 ^#^	4.20 ± 1.02	3.39 ± 0.65 *	4.90 ± 1.94
Neutrophils (×10^4^ cells)	5.41 ± 1.14	53.09 ± 6.44 ^###^	21.19 ± 5.56 **	11.53 ± 2.39 ***	23.18 ± 5.37 **
Eosinophils (×10^4^ cells)	11.74 ± 3.89	48.90 ± 12.93 ^##^	44.74 ± 16.68	28.97 ± 3.16	47.26 ± 9.15
CD4^+^ (×10^4^ cells)	0.55 ± 0.25	31.56 ± 8.11 ^##^	10.51 ± 2.32 *	7.71 ± 1.00 **	15.90 ± 4.20
CD8^+^ (×10^4^ cells)	0.10 ± 0.06	17.01 ± 2.15 ^###^	3.98 ± 1.01 **	6.69 ± 2.30 *	10.39 ± 3.02
CD62L^−^/CD44^high+^ (×10^4^ cells)	1.64 ± 0.39	89.75 ± 15.92 ^###^	55.28 ± 15.35	25.55 ± 3.39 *	56.39 ± 10.65
Gr-1^+^SiglecF^−^ (×10^4^ cells)	1.09 ± 0.53	53.18 ± 9.61 ^##^	14.35 ± 4.32 *	7.63 ± 2.02 **	20.40 ± 4.99 *
Lung					
Lymphocytes (×10^4^ cells)	14.46 ± 1.96	24.06 ± 4.26 ^#^	33.31 ± 2.46	29.20 ± 9.50	37.82 ± 12.24
Neutrophils (×10^4^ cells)	24.15 ± 5.70	81.34 ± 17.00 ^##^	50.86 ± 2.19	40.57 ± 10.67 *	51.09 ± 10.56
Eosinophils (×10^4^ cells)	5.96 ± 0.73	12.20 ± 2.48 ^#^	12.53 ± 0.55	12.55 ± 4.09	12.70 ± 3.01
CD4^+^ (×10^4^ cells)	15.94 ± 3.51	34.79 ± 5.47 ^##^	33.93 ± 1.28	30.73 ± 9.48	32.90 ± 7.49
CD8^+^ (×10^4^ cells)	6.68 ± 1.49	21.77 ± 4.24 ^##^	17.45 ± 0.79	14.79 ± 3.55	15.70 ± 4.68
CD4^+^ CD69^+^ (×10^4^ cells)	1.38 ± 0.44	4.06 ± 0.67 ^##^	2.41 ± 0.34 *	2.13 ± 0.59 *	2.66 ± 0.59
CD62L^−^/CD44^high+^ (×10^4^ cells)	3.88 ± 0.76	17.88 ± 1.47 ^###^	9.00 ± 1.01 **	9.68 ± 2.79 *	11.82 ± 3.51
Gr-1^+^SiglecF^−^ (×10^4^ cells)	9.49 ± 2.76	41.83 ± 3.15 ^##^	19.26 ± 2.26 **	13.45 ± 2.52 **	21.31 ± 3.69 *
Gr-1^+^CD11b^+^ (×10^4^ cells)	15.58 ± 3.13	56.92 ± 7.92 ^##^	28.78 ± 2.15 *	21.39 ± 3.97 *	30.03 ± 6.30
CD21^+^/CD35^+^B220^+^ (×10^4^ cells)	4.33 ± 2.15	21.08 ± 4.84 ^##^	8.09 ± 2.14 *	10.40 ± 3.27	14.78 ± 5.39

Abbreviations: NC, normal control; CTL, control; Dexa, dexamethasone; CD, cluster of differentiation; B220, B cell specific isoform of CD45; Gr, granulocytes; CD62L, leucocyte endothelial cellular adhesion molecule; SiglecF, sialic-acid-binding immunoglobulin-like lectin F. The data are presented as the mean ± SEM (*n* = 8/group). ^#^ *p* < 0.05, ^##^ *p* < 0.01, ^###^ *p* < 0.001 vs. the NC group, and * *p* < 0.05, ** *p* < 0.01, *** *p* < 0.001 vs. the CTL group.

## Data Availability

The data presented in this study are available in the article.

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
