# Peer review of "Siraitia grosvenorii Extract Attenuates Airway Inflammation in a Mouse Model of Respiratory Disease Induced by Particulate Matter 10 Plus Diesel Exhaust Particles"

_nutrients, 2023, doi:10.3390/nu15194140_

Round 1

Reviewer 1 Report

Overall, the manuscript needs to be rewritten as the current structure and flow is not optimal. The results are confusing to follow, and the Discussion lacks discussion. Exact P-values and quantitative measurement for histology is required. Methods are not clear, e.g., sample size. Below are some major concerns to be addressed:

Abstract:

1) More background information about SGE is required and why did you decide to investigate it.

2) Some methods are required in the abstract. The abstract now is just full of results statements.

3) Overall, the abstract flow and structure needs to be rewritten to improve readability.

Introduction:

1) Line 40-45: this is just a list of what is in PM/DEP. It will be more useful to reference studies showing how PM/DEP affects lung health.

2) Line 51: some elaboration is required about its properties.

3)Justification and novelty of the study needs to be stronger.

Methods:

1) Line 65: A brief line on its preparation is required.

2) Figure 1A: can be improved by showing all 5 groups rather than just one line.

3) Never start a sentence with an abbreviation. Please check throughout the manuscript.

4) Sample size needs to be stated clearly in-text.

5) Why were male mice chosen and not female mice?

6) A lot of abbreviations were not defined in the first use, i.e., flow cytometry section.

7) Section 2.6: was this from another set of mice? Or the same set? The left or right lung was used for histology? Was the person that perform the measurements blinded to the groups?

8) Table 1: The genes must be defined in the abbreviation list.

9) Western blot: Concentration of the antibodies used must be listed. Protein names must be defined in its first use. Were the phosphorylated proteins detected on the same blot as the total protein? Or were they on separate blots?

Results:

1) Never start a sentence with an abbreviation. Please check throughout the manuscript.

2) Exact P-values should be written in-text. Even non-significant P-values should be stated.

3) ‘Normal’ is not the right term to use. Please rename this group.

4) Figure 1C is missing.

5) Line 159: the ‘B’ in ‘blood’ doesn’t need to be in capital letter.

6) Section 3.4: is there any quantitative measurements? Qualitative measurements are not sufficient. Cannot just be eyeballing the slides.

7) The flow of the Results doesn’t match Methods. Please revise.

8) Table 2: Must define protein names in the abbreviation list.

9) Figure 6: were all the proteins performed on the same blot? Otherwise, there should be more than one actin band.

10)   Sample size unclear for all experiments.

Discussion:

1) Never start a sentence with an abbreviation.

2) What are the limitations of the study and what are the novelty about this study?

3) Any comment about sex difference? What would be the practical dose and frequency of SGE is required for human? More discussion on clinical impact is required.

4) Overall, the Discussion needs to be rewritten as the Discussion as it is currently poorly written. It is now written as one line about a protein/gene and then what the result of this manuscript is with many short paragraphs, therefore needs to be rewritten/restructure to improve readability.

Need extensive editing to improve readability.

Reviewer 2 Report

In the work entitled “Siraitia grosvenorii Extract Attenuates Airway Inflammation in a Mouse Model of Respiratory Disease Induced by Particulate Matter 10 plus Diesel Exhaust Particles”, the authors evaluated the inhibitory effect of Siraitia grosvenorii extract on airway inflammation in mice exposed to a fine dust mixture of PM10 and PM10D. The manuscript is well organized with necessary data and within the scope of this Journal. The introduction and background are reasonable given the promise of the paper. Figures and tables are comprehensive and helpful.My comments regarding this article are as follows,

1.       Some English grammar and spelling corrections must be done throughout the whole paper.

2.       I strongly recommend to the authors to use most recently published articles in their paper, for example there are several published papers about bioactive composition of Siraitia grosvenorii and their biological activities, however they are not included in the paper. Some articles are listed in the below:

# https://doi.org/10.1016/j.jfca.2022.105070

https://doi.org/10.1016/j.foodchem.2022.133593

3.       Which plant tissue did the authors use for extraction? Has there been an extract from the fruit of Siraitia grosvenorii? If yes, it is better to use Siraitia grosvenorii fruit extract instead of Siraitia grosvenorii extract in the whole text including title. Use SGFE instead of SGE.

4.       In the Figure 2 caption, the extra E should be removed at the beginning of line 169.

5.       Write in third person, avoid personal pronouns, such as “we” and “our” lines 245, 260, 288…

6.       It is better to authors provide a schematic representation for signaling pathways involved in the reduction of airway inflammation in a mouse model of respiratory disease

7.       Gene names should be italic throughout the manuscript including tables

8.       The references cited are in some instances old (references 2, 3, 8, 20, 21,), it is of utmost importance to cite newest possible references, and not the ones that are 20-25 or 30 years old. Majority of cited publications must be published in the last 10 years meaning from 2013 to 2023. Exchange the old references with new ones.

Minor editing of English language required

Round 2

Reviewer 1 Report

This manuscript has improved but there are still some minor corrections required.

1) Please check your symbols, in particular for alpha and beta.

2) he histology assessment is considered as ‘semi-quantitative’ and not ‘quantitative’.

3) Some of the exact P-values are still not stated.

4) Dot plots is required instead of bar graphs so readers can see the variability between the 8 mice in each group.

5) Is the left lung used for flow cytometry, qPCR and Western blot? Are they all from the same mice?

6) here exactly were the mice kept (facility name)? How were the mice euthanised?

7) Western blot: “the antibodies raised against beta-actin” is incorrect. Rather you need to say that the proteins will be analysed relative to beta-actin at the end.

8) Western blot: how much proteins were loaded onto the gel? Did you check they are the same amount across the wells?

9) Another limitation of this study is that there is no lung function and airway morphometry data.

This version has improved but there are still important methodology and results details that are required before this manuscript can be accepted.

Reviewer 2 Report

Accept

Author Response

Thank you so much about 'accept'.